# Morphogenesis Changes in Protocorm Development during Symbiotic Seed Germination of *Dendrobium chrysotoxum* (Orchidaceae) with Its Mycobiont, *Tulasnella* sp.

Xinzhen Gao [1,2], Yu Wang [1,3], Die Deng [4], Yinling Luo [4], Shicheng Shao [5,*] and Yan Luo [1,*]

1. Southeast Asia Biodiversity Research Institute, Chinese Academy of Sciences & Center for Integrative Conservation, Xishuangbanna Tropical Botanical Garden, Chinese Academy of Sciences, Mengla 666303, China
2. University of Chinese Academy of Sciences, Beijing 100049, China
3. School of Landscape Architecture, Southwest Forestry University, Kunming 650224, China
4. College of Biology and Chemistry, Puer University, Puer 665000, China
5. CAS Key Laboratory of Tropical Forest Ecology, Xishuangbanna Tropical Botanical Garden, Chinese Academy of Sciences, Mengla 666303, China
* Correspondence: shaoshicheng@xtbg.org.cn (S.S.); luoyan@xtbg.org.cn (Y.L.); Tel.: +86-189-8819-6796 (S.S.); +86-135-7810-9629 (Y.L.)

**Abstract:** The endangered epiphytic orchid, *Dendrobium chrysotoxum*, is known for its ornamental and medicinal uses. However, knowledge of this orchid's symbiotic seed germination, protocorm anatomy, and developmental morphology is completely unknown. In this study, we investigated the process of protocorm development of *D. chrysotoxum* during symbiotic germination using anatomical technologies and scanning electron microscopy. There are six development stages that were morphologically and anatomically defined during symbiotic seed germination. The embryo transformed into a protocorm at stage two, and a protrusion developed from the top of the protocorm at stage three and elongated to form the cotyledon at stage four. The stem apical meristem (SAM) was initiated at stage three and well developed at stage four. The first leaf and the root appeared at stages five and six, respectively. The hyphae entered through the micropylar end of the seed at stage one and then invaded the protocorm through rhizoids when rhizoids formed. Invading fungal hyphae colonized the inner cortex at the base of protocorms, formed pelotons, and were digested by host cells later. We conclude that protocorm development is programmed by the embryo, which determines the structure and function of the protocorm. The two developmental zones in a polarized *D. chrysotoxum* embryo include the smaller cells zone, which forms the cotyledon and a shoot apical meristem at the apical end, and the larger cells zone, which forms the mycorrhiza to house the symbiont at the basal end. These results will provide important insights for further research on the mechanisms underlying orchid-fungi symbiosis and enhance the understanding of orchid evolution.

**Keywords:** *Dendrobium chrysotoxum*; anatomy; protocorm development; symbiotic seed germination; mycorrhizal association; Orchidaceae

## 1. Introduction

Most orchid species have small, less highly developed seeds consisting of seed coats and rudimentary embryos [1,2]. Due to the lack of endosperm that is normally required as an initial energy resource, orchid seeds cannot germinate on their own under natural conditions. Bernard first demonstrated the dependency on fungi during seed germination in *Neottia* in 1899 [3,4]. The fungal mycobionts can supply nutrients to promote seed germination [5]. The presence of compatible mycorrhizal fungi plays a key role in the success of seed germination. Orchid embryos are protoembryos with developmental programs more complex than other flowering plants [6]. It is essential to form protocorms prior to seedling formation during seed germination in orchids. Protocorm formation is a

characteristic feature of orchid seed germination. The protocorm is thought to be a unique structure in which symbiotic associations with mycorrhizal fungi are established and shoot apical meristems are formed [7]. A successful establishment with an appropriate symbiont will lead to seedling formation [5].

Numerous studies on seed germination in temperate terrestrial orchids have resulted in a considerable amount of information on seedling establishment and fungi interaction with protocorms [8–11]. Orchid seeds generally germinate through a series of stages: first, the embryo within the seed coat will enlarge, then develop into a protocorm and resume mitotic activities, especially at the future shoot pole, followed by seedling formation with both a shoot and roots [7,12].

The unique structure and function of the protocorm have led to great controversy about its structure and properties. Protocorm development is considered an intermediate stage between the undifferentiated embryo and the seedling [2]. Some researchers regard protocorm as the extension of embryonic development [2,13], while others consider protocorm more similar to underdeveloped seedlings [14,15]. In many species, a foliaceous or crest-like protrusion forms on the flat upper surface of protocorms [16,17], but what this structure should be referred to has caused much controversy. The protrusion on the protocorms was described as the first leaf in many species, such as *Platanthera clavellata* [18], *Paphiopedilum wardii* [19], etc. On the other hand, it could be assumed to be a cotyledon in orchids such as *Dendrobium huoshanense* [20] and *Dendrobium moniliforme* [21]. There are several interpretations suggested for this organ, but none have been accepted as conclusive. Moreover, the process of tissue differentiation in protocorms and their physiological functions during orchid seed germination are not well understood.

Detailed reports of morphogenesis changes during symbiotic seed germination in orchids are still limited. Using transmission electron microscopy, previous studies have focused on significant morphological changes in fungal hyphae in colony cells during symbiotic germination [8,9,22–24]. Richardson et al. [9] observed that fungal hyphae invaded the embryo of *Platanthera hyperborea* through the suspensor and formed pelotons thereafter. Uetake et al. [23] investigated the cellular and fungal hyphae ultrastructural changes in *Spiranthes sinensis* during different germination stages and found pelotons digested in the inner cortical parenchyma at the first stage. Fungal hyphae will repeatedly infect cells during all growth stages, and cells containing clumped hyphae will frequently be recolonized by viable hyphae [8]. The developmental process of symbiotic germination is still less known, thereby limiting our understanding of mycorrhizal fungi-plant interactions. Detailed morphological changes in the symbiotic seed germination of orchids need to be investigated.

*Dendrobium chrysotoxum* Lindl, an endangered (sub) tropical epiphytic orchid, is distributed in Southwestern China, Northeastern India, and Southeast Asian countries such as Laos, Thailand, and Vietnam [25]. In traditional Chinese medicine, the stem of *D. chrysotoxum* contains a variety of bioactive materials, such as polysaccharides, polyphenols, and alkaloids, which confer its high medicinal value as an antipyretic, analgesic, and immunomodulating agent [26–28]. Furthermore, *D. chrysotoxum* is an ornamental plant due to its bright yellow color, long flowering period, and long-lasting floral scent [29]. In our previous study, a *Tulasnella* fungal strain (GC-15) was isolated from *D. chrysotoxum* protocorms using an in situ seed baiting technique, which is significantly effective at promoting in vitro seed germination and protocorm development [30]. However, the interaction between *D. chrysotoxum* protocorm and its mycobiont during seed germination is still unclear.

In the present study, we investigated detailed morphogenesis changes during the symbiotic germination of *D. chrysotoxum* inoculated with *Tulasnella* sp. using anatomical technologies and scanning electron microscopy (SEM). The aim of our study was to clarify morphological and anatomical changes during protocorm development in order to better understand the mutualism between *D. chrysotoxum* and its mycobiont. This study provided new insights into the structure and function of epiphytic orchid protocorms and shed light on the symbiont relationship between orchids and fungi.

## 2. Materials and Methods

### 2.1. Seed Collection

*Dendrobium chrysotoxum* was grown in the collection gardens of the Xishuangbanna Tropical Botanical Garden (XTBG hereafter), Chinese Academy of Sciences, Yunnan Province, China (Figure 1a, 21°54′ N, 101°46′ E, 580 m in altitude). Mature capsules were harvested prior to dehiscence in March 2021 (Figure 1b). Capsules were sterilized on a surface with 75% ethanol. Seeds were carefully removed from the capsules and then dried for 4 days using anhydrous calcium chloride (CaCl$_2$) in airtight glass containers. Finally, seeds were put into a 1.5 mL centrifuge tube and stored at −20 °C until use.

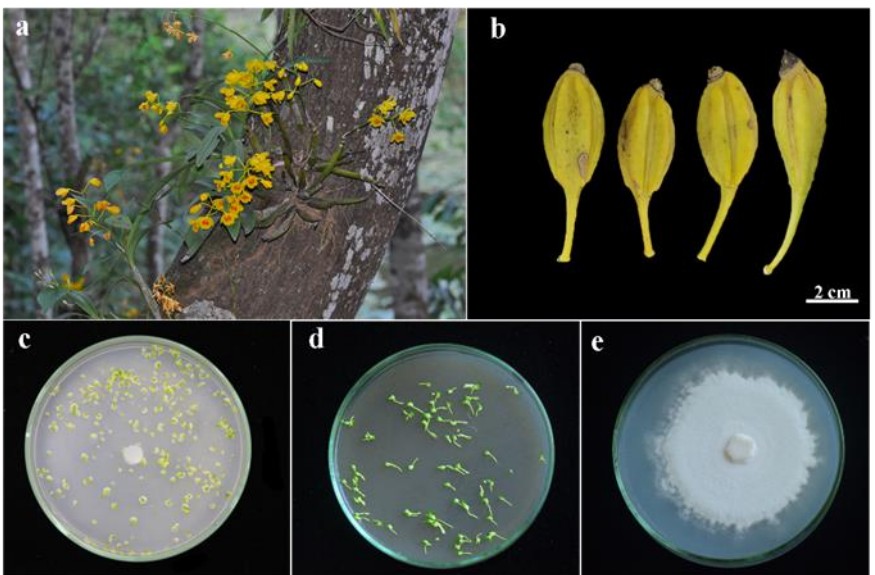

**Figure 1.** Materials and culture methods. (**a**) Flowering *Dendrobium chrysotoxum* as an epiphytic orchid; (**b**) Mature capsules of *Dendrobium chrysotoxum*; (**c**) Symbiotic germination system; (**d**) Asymbiotic germination system; (**e**) Culture of fungal strain GC-15 (*Tulasnella* sp.) on PDA medium.

### 2.2. Fungal Culture

The strain *Tulasnella* sp. (GC-15) with promotion on seed germination was used in this study [30]. The strain was deposited in XTBG, Yunnan, China. Before symbiotic germination, the strain was cultured on potato dextrose agar medium (PDA: potato 200 g/L, glucose 20 g/L, agar 15 g/L) in darkness at 25 ± 1 °C for 7–9 days. The actively growing mycelium on the colony margin was used for fungal inoculation (Figure 1e).

### 2.3. In Vitro Seed Germination

Asymbiotic germination (Figure 1d): Seeds were sterilized in a 1% NaClO solution for 3 min and rinsed three times with sterile water. After surface sterilization, seeds were sown evenly on the medium in a 9 cm Petri dish and finally sealed with paraffin.

The culture medium used in asymbiotic germination was 1/2 Murashige and Skoog (MS) medium, supplemented with 20 g/L sucrose, 4 g/L agar, 40 g/L bananas, 60 g/L potatoes, and 1 g/L activated charcoal. The medium's pH was adjusted to 5.8 before autoclaving at 121 °C for 15 min. A total of 40 plates were incubated at 23 ± 2 °C under a 12 h light/12 h dark photoperiod with cool-white, fluorescent lamps, providing a light intensity of 1800 lx.

Symbiotic germination (Figure 1c): Seeds were sterilized and sown using the methods described above. A fungal plug with a 0.5 cm$^3$ volume of actively growing mycelium or a PDA plug without fungi (as a control) was placed on the center of oatmeal agar medium (OMA: 4 g/L rolled oats, 8 g/L agar) [30]. After seeds were sown evenly around the plug and sealed up, a total of 40 plates were incubated under the same conditions as mentioned above.

Seed germination and protocorm development were monitored and recorded every 15 days (15, 30, 45, 60, and 75 d) after seed sowing until the seedling was established, using non-contaminated plates. Seed germination rate and percentage of developed protocorms (stages 1-6) were calculated by dividing the number of seeds in each developmental stage after incubation by the total number of germinated seeds [31]. The percentage of germination (G) = (S1 + S2 + S3 + S4 + S5 + S6)/(S0 + S1 + S2 + S3 + S4 + S5 + S6), where S0 is the number of seeds in stage 0, and so on. Seeds without embryos were not counted. Each treatment consisted of five replicates.

### 2.4. Microscopic Morphological Observations

Ungerminated seeds, a large number of protocorms at various developmental stages, and seedlings from each medium were sampled for microscopic investigations. At least five samples at each stage were photographed using a stereoscope (Nikon, SMZ800N). For SEM (Carl Zeiss EVO LS10) observations, the protocorms and seedlings were dehydrated with grade alcohol series at 50%, 70%, 85%, 95%, and 100% for 15–20 min. Then the samples were critical-point dried and sputter-coated with gold before being mounted on stubs and examined under SEM. Seed and embryo (N = 30) length or width were measured by Image J. Embryo occupation and free air space were calculated according to the methods provided by Arditti and Ghani [32]:

$$\text{Seed volume (SV)} = 2(\pi/3 \; r^2 h) \tag{1}$$

$$\text{Embryo volume (EV)} = 4/3 \; \pi a b^2 \tag{2}$$

$$\text{Embryo occupation (EO)} = \text{EV}/\text{SV} \cdot 100\% \tag{3}$$

$$\text{Air space (AS)} = 1 - \text{EO} \tag{4}$$

(r: 1/2 seed width; h: 1/2 seed length; a: 1/2 embryo length; b: 1/2 embryo width)

### 2.5. Histological and Histochemical Studies

Different stages of protocorms were collected and fixed in FAA for histological and histochemical studies. Samples were dehydrated using a graded ethanol series of 50%, 70%, 85%, 95%, and 100% for 30 min. Next, samples were infiltrated in an ethanol-xylene mixture (1:1; *v/v*), followed by two changes of pure xylene. Paraffin was added to the xylene solution until saturated at a temperature gradient (37 °C, 42 °C, and 57 °C). Then transferred the samples into pure paraffin. After incubation in pure paraffin overnight, samples were embedded in paraffin. The wax was sliced into 5 μm thickness on an Autocut rotary microtome (Leica). After rehydration with an ethanol series, the slides were stained with hematoxylin dye for 10 min, rinsed with running water for 10 min, and dehydrated using a graded ethanol series. Finally, the slides were transparentized in xylene and coverslipped with neutral resin before being photographed by microscopy (NICON ECLIPSE Ci-L, Tokyo, Japan). More than 5 samples at each stage were prepared into paraffin sections. In order to observe the distribution of polysaccharides and proteins, the slides were stained with Periodic acid-Schiff dye solution (PAS) for 5 min and counterstained with naphthol yellow S for 5 min. Polysaccharides were stained purplish red by PAS, and proteins were stained yellow by Naphthol yellow S.

### 2.6. Fungal Infection Observation

Trypan blue staining was used for observing fungal invasion following the methods described by Ma et al. [33] with a few modifications. Protocorms were immersed in a 10% potassium hydroxide (KOH) solution at 90 °C for 40 min and bleached with a 30% hydrogen peroxide solution for 30 min. Then the sample was stained with 0.05% (*w/v*) trypan blue in a lactic acid glycerol solution for 8 min and decolored in an acetic glycerol

solution for 20 min. The fungal hyphae were observed using microscopy (NIKON ECLIPSE Ci-L, Tokyo, Japan).

*2.7. Statistical Analysis*

To compare the effect of seed germination between symbiotic and asymbiotic cultures, the data for each developmental stage were analyzed using an analysis of variance followed by an independent-sample *t* test ($\alpha$ = 0.05). Statistical analysis was performed using SPSS 26.0, and the results were graphed by Origin 2023.

## 3. Results

*3.1. The Seed Characteristics of Dendrobium chrysotoxum*

The dehydrated seeds were ellipsoid and measured 392.97 $\pm$ 41.1 $\times$ 101.52 $\pm$ 10.5 μm in size; the embryos were golden yellow in color and measured 224.01 $\pm$ 19.83 $\times$ 99.18 $\pm$ 9.6 μm in size (Table 1; Figure 2a,e). The transparent testa surrounding embryos was highly elongated (three to five cells along the longitudinal axis), forming a distinct marginal ridge with lignin strips or bars on the surface (Figure 2f,g). The apical (chalazal) end of the testa was acute, while the basal (micropylar) end of the testa was truncate (Figure 2d).

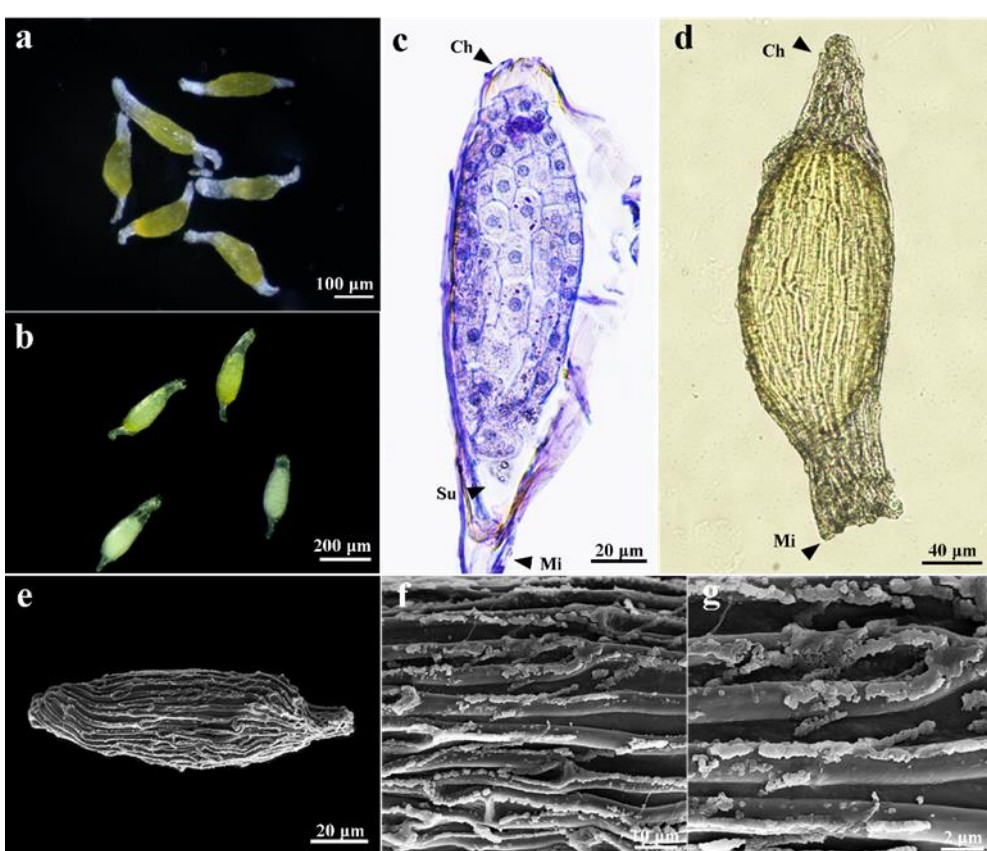

**Figure 2.** The seed characteristics of *Dendrobium chrysotoxum*. (**a**) Dehydrated seeds; (**b**) Hydrated seeds; (**c**) Longitudinal sections of seeds showing a gradient of cell size with small cells located on the chalazal end (Ch) and large cells on the micropylar end (Mi), a degenerated suspensor cell (Su) at the base of the embryo; (**d**) Light microscopy photograph of dehydrated seeds. Ch: chalazal end; Mi: micropylar end; (**e–g**) SEM photographs of dehydrated seeds.

**Table 1.** Seed features of *Dendrobium chrysotoxum* (±standard deviation).

| Seed Characteristics | |
| --- | --- |
| Seed length (μm) | 392.97 ± 41.1 |
| Seed width (μm) | 101.52 ± 10.5 |
| Seed L/W ratio | 3.89 ± 0.38 |
| Embryo length (μm) | 224.01 ± 19.83 |
| Embryo width (μm) | 99.18 ± 9.6 |
| Free air space (%) | 33.4 ± 0.07 |
| Embryo occupation (%) | 66.6 ± 0.07 |

As seeds matured, embryo polarity was established. The embryo cells of *D. chrysotoxum* have dense cytoplasm and two recognized regions in terms of cell size and form. Smaller cells with conspicuous nuclei located at the apical (chalazal) end and large and vacuolated cells located at the basal (micropylar) end of the embryo indicate structural polarization in the embryo of *D. chrysotoxum* (Figure 2c). A single, small, degenerate suspensor cell located at the bottom of the embryo near the micropylar end (Figure 2c).

*3.2. Seed Germination and Protocorm Development*

Based on morphological and histological analysis, six developmental stages were defined. At stage 1, seed imbibition was observed after 1 d of seed sowing (Figure 2b,d), and embryos had become swollen soon after 3–5 d of culture (Figure 3a). Upper cells are small, while lower cells are enlarged and vacuolated (Figure 4a). At stage 2, embryos had enlarged rapidly, resulting in testa rupture by 13–15 d (Figure 3b). Embryos initially developed into globular protocorms and continued to increase in diameter, forming light green spherical protocorms (Figure 3b). There was a hardly visible protrusion on the flat upper part of the protocorm. The protomeristems occurred at the upper part of the protocorm, as characterized by a couple of small cells with conspicuous nuclei when compared to neighboring cells (Figure 4b,e). Hyphae of *Tulasnella* sp. (GC-15) penetrated seeds through the micropyle and entered the embryo through the degenerated suspensor, exhibiting the capacity to infect seeds and form pelotons within larger parenchymatous cells at the basal end of the protocorm (Figure 4b,l,m). The formation of pelotons indicate the success of mycorrhizal establishment.

At stage 3, the protocorms increased in size gradually to form oblate spheroids by 18–20 d (Figure 3c). The protrusion on the apical surface of protocorms continued to grow and elongate (Figure 3c). As protrusion elongated, the protomeristematic cells at the center of the protocorm (but above the basal mycorrhizal zone) became meristematic and were small, square to polygon shaped, with a large nucleus to cytoplasm ratio, eventually giving rise to the shoot apical meristem (SAM) initials (Figure 4c). A cleft appeared between SAM and surrounding cells (Figure 4f). These cytological features clearly mark the initiation of a SAM. Numerous rhizoids in clusters developed along the base of the protocorm (Figure 5b,c). Rhizoids were formed by the extension of clusters of epidermal cells in protocorms (Figure 5b). Hyphae invasion through rhizoids could be observed at this stage (Figures 4n and 5c). Large parenchymatous cells with pelotons in different forms occupied the lower part of the protocorm (Figure 5h–j).

At stage 4, protrusions on the apical part of the protocorm elongated and became highly chlorophyllous by 25 d (Figures 3d and 5d). Stomata appeared on the surface of the protrusion, and irregular-shaped parenchyma cells with abundant chloroplasts appeared inside (Figures 5d and 6f). It was evident that these anatomical features of protrusion belonged to the foliage organ. This protrusion of protocorm developed from embryonic cells, not from a SAM; therefore, it should be considered a cotyledon rather than the first true leaf. Histological sectioning revealed that a small depression appeared on the initiation of SAM, clearly indicating the SAM was located at the center of the protocorm and below the cotyledon (Figure 4g). Soon after, a fully developed SAM bearing leaf primordia was enclosed within the protocorm (Figure 4d,h,i).



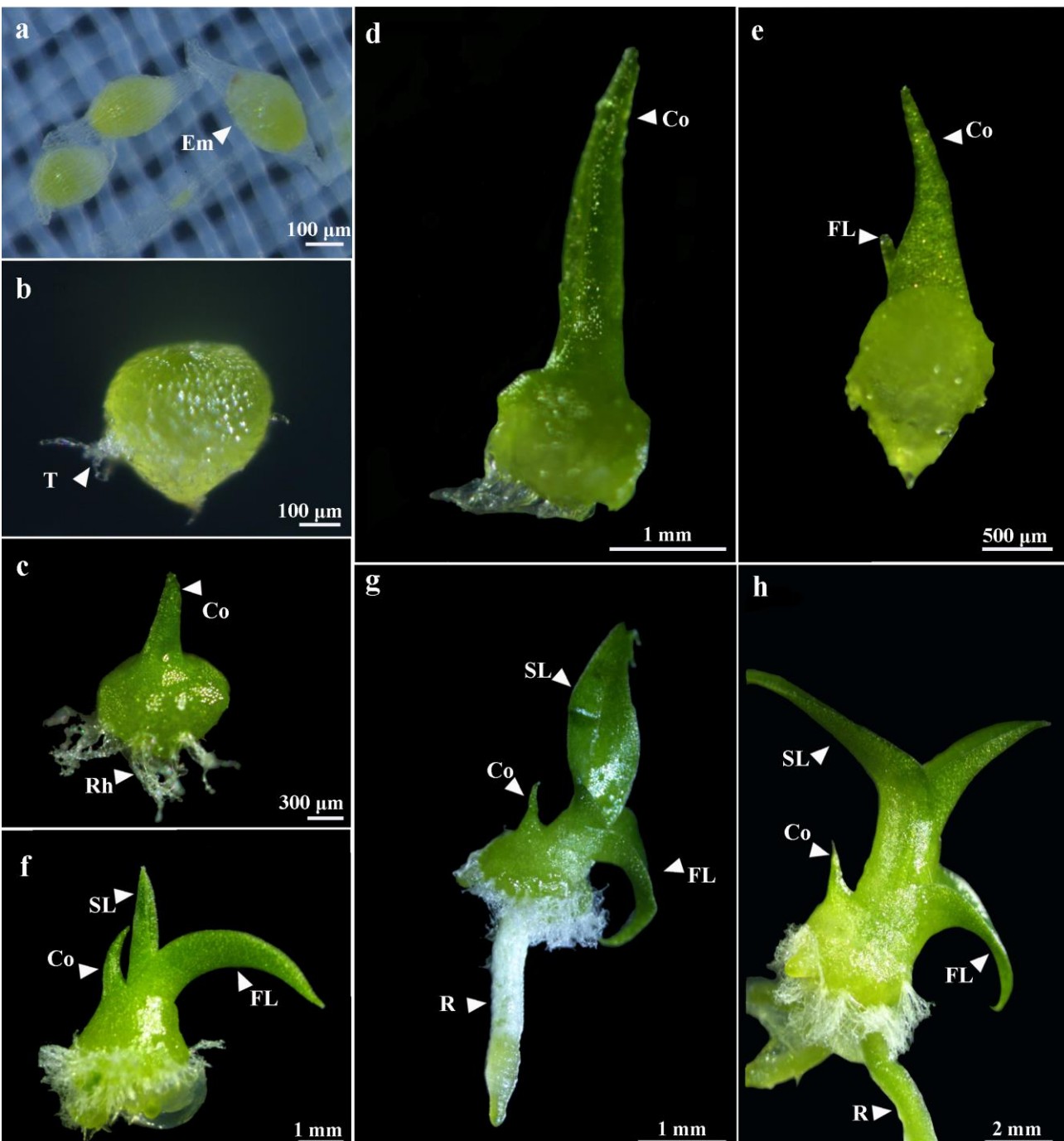

**Figure 3.** Developmental stages of symbiotic seed germination of *Dendrobium chrysotoxum*. (**a**) Stage 1 embryo (Em) swollen; (**b**) Stage 2 embryo enlarged and testa (T) ruptured; (**c**) Stage 3 appearance of cotyledon (Co) and rhizoids (Rh); (**d**) Stage 4 continued elongation of the cotyledon (Co); (**e**,**f**) Stage 5 appearance of the first (FL) and the second leaf (SL). Co: Cotyledon; (**g**,**h**) Stage 6 seedling with cotyledon, true leaves, and roots (R). Co: Cotyledon; FL: First leaf; SL: Second leaf.

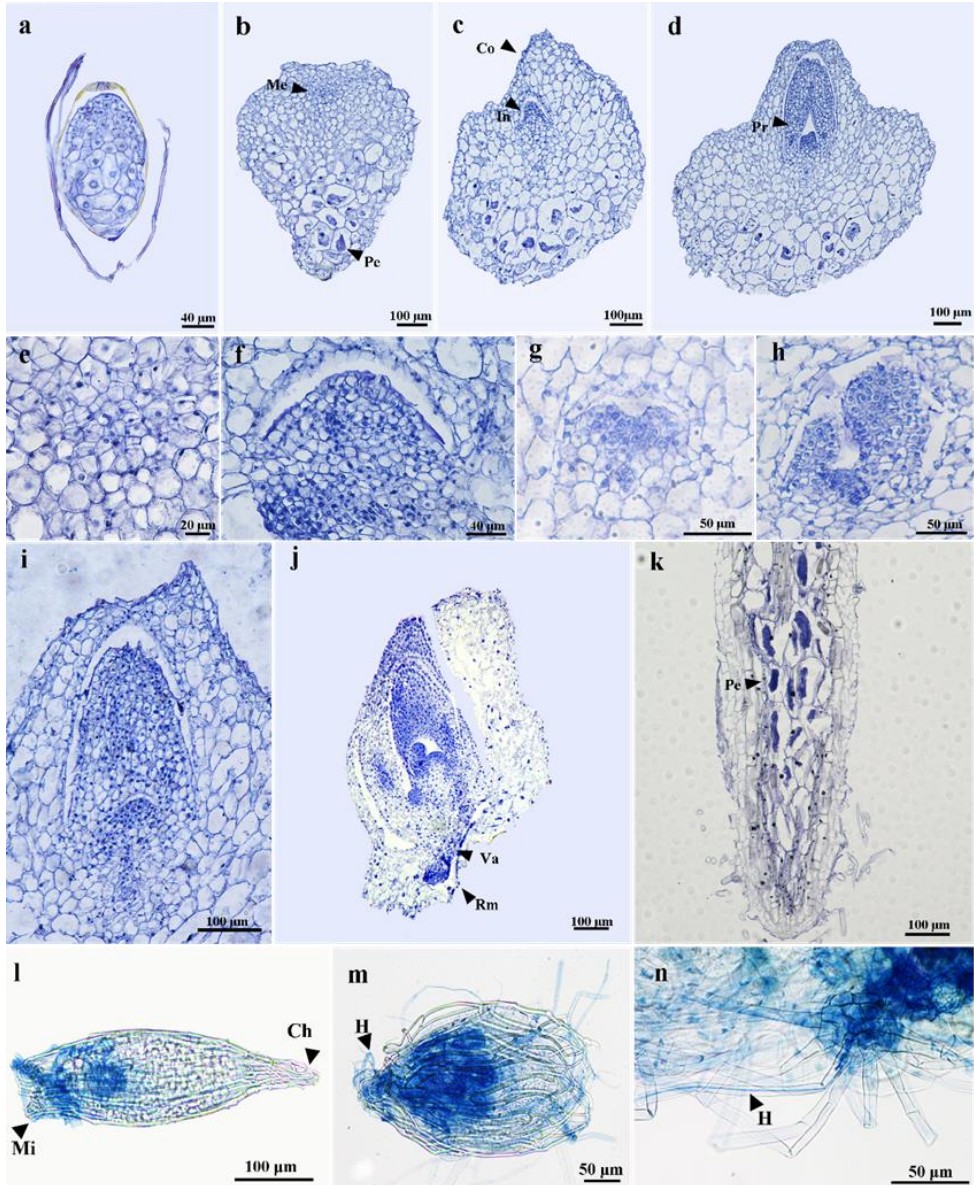

**Figure 4.** Longitudinal sections (**a–k**) and trypan blue stained (**l–n**) *Dendrobium chrysotoxum* embryos and protocorms at different developmental stages. (**a**) Stage 1: swollen embryo showing small cells at the chalazal end and big ones at the micropylar end; (**b**) Stage 2: young protocorm showing the protomeristem (Me) formation at the top, and peloton (Pe) formation at the base; (**c**) Stage 3: the cotyledon (Co) developing from the apical surface of the protocorm and the stem apical meristem (SAM) initials (In) emerging; (**d**) Stage 4: well-developed SAM with leaf primordium (Pr); (**e**) The protomeristematic cells at the central of the protocorm at stage 2; (**f**) The SAM initials formation at stage 3; (**g–i**) The SAM continues to grow in size, surrounded by developing leaf primordia at stage 4; (**j**) The root (Rm) primordium developed from the basal surface of the SAM and a vascular (Va) connected root meristem and shoot meristem at stage 5; (**k**) Seedling root at stage 6 with fungal peloton (Pe); (**l**) Imbibed seed stained in trypan blue at stage 1, fungal hyphae invade seed from micropyle end (Mi). Ch: chalazal end; (**m**) Fungal hyphae (H) colonized to form pelotons at base of protocorm at stage 2; (**n**) Fungal hyphae (H) invaded protocorm from rhizoids.

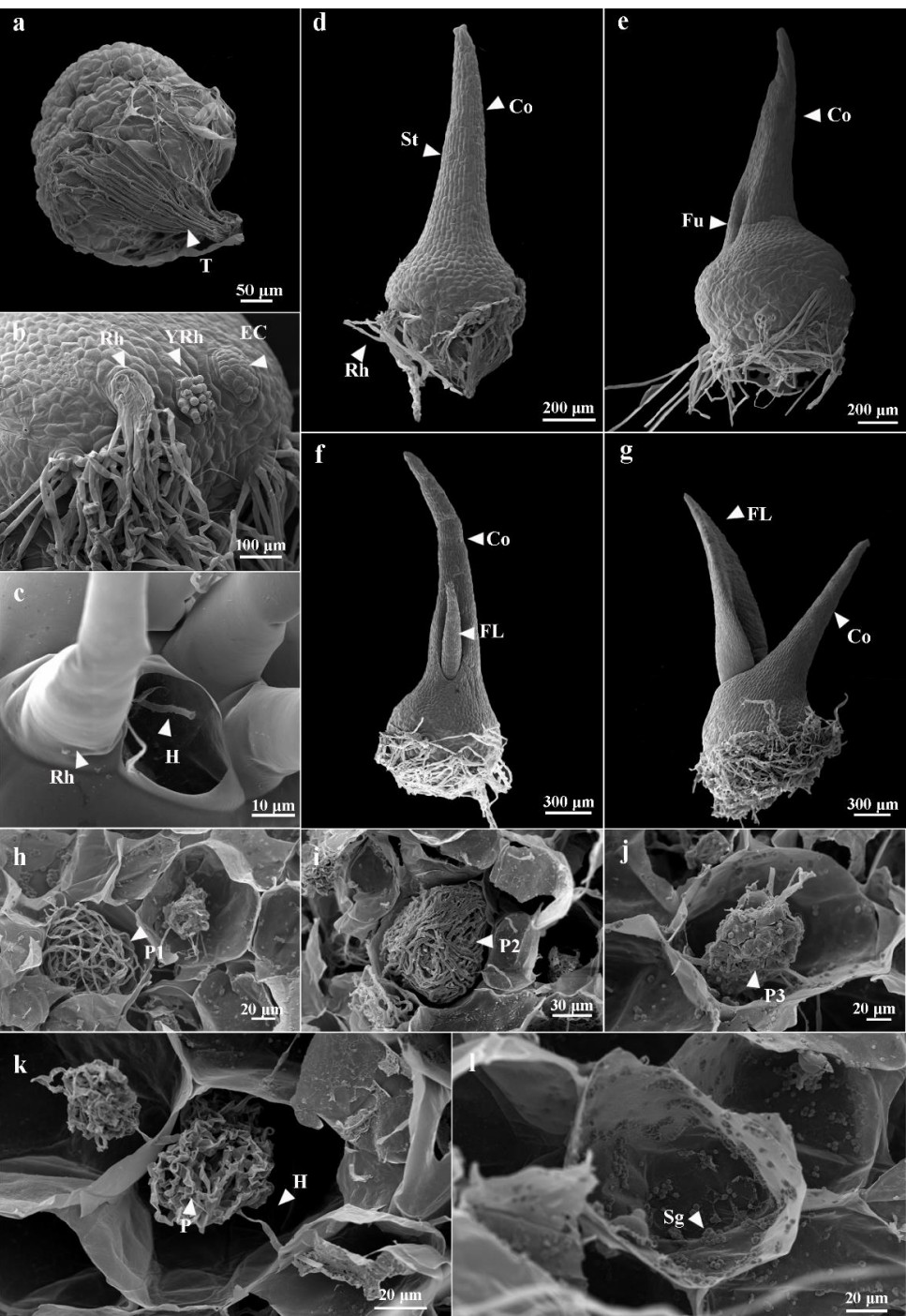

**Figure 5.** SEM photographs of protocorms of *Dendrobium chrysotoxum* at different developmental stages (**a**–**g**) and various pelotons within cortex cells (**h**–**l**). (**a**) Stage 2: young protocorms with ruptured testa (T); (**b**) Stage 3: epidermic cell clusters (EC), young rhizoids (YRh), and elongated rhizoids (Rh) emerging at the base of the protocorm; (**c**) Stage 3: fungal hyphae (H) penetrating through the rhizoid (Rh); (**d**) Stage 4: cotyledon (Co) formation with stomata (St) appearing on the surface. Rh: rhizoid; (**e**) Stage 5: a furrow (Fu) emerging at the base of cotyledon (Co); (**f**,**g**) Stage 5: the first leaf (FL) emerging. Co: Cotyledon; (**h**) Loose pelotons (P1), fungal hyphae were intact and healthy; (**i**) Continued tightening of the pelotons (P2); (**j**) Clumped pelotons (P3), fungal hyphae were digested and collapsed; (**k**) Pelotons (P) in adjacent cells connected by fungal hyphae (H); (**l**) Cortex cells without pelotons but containing some starch granules (Sg).

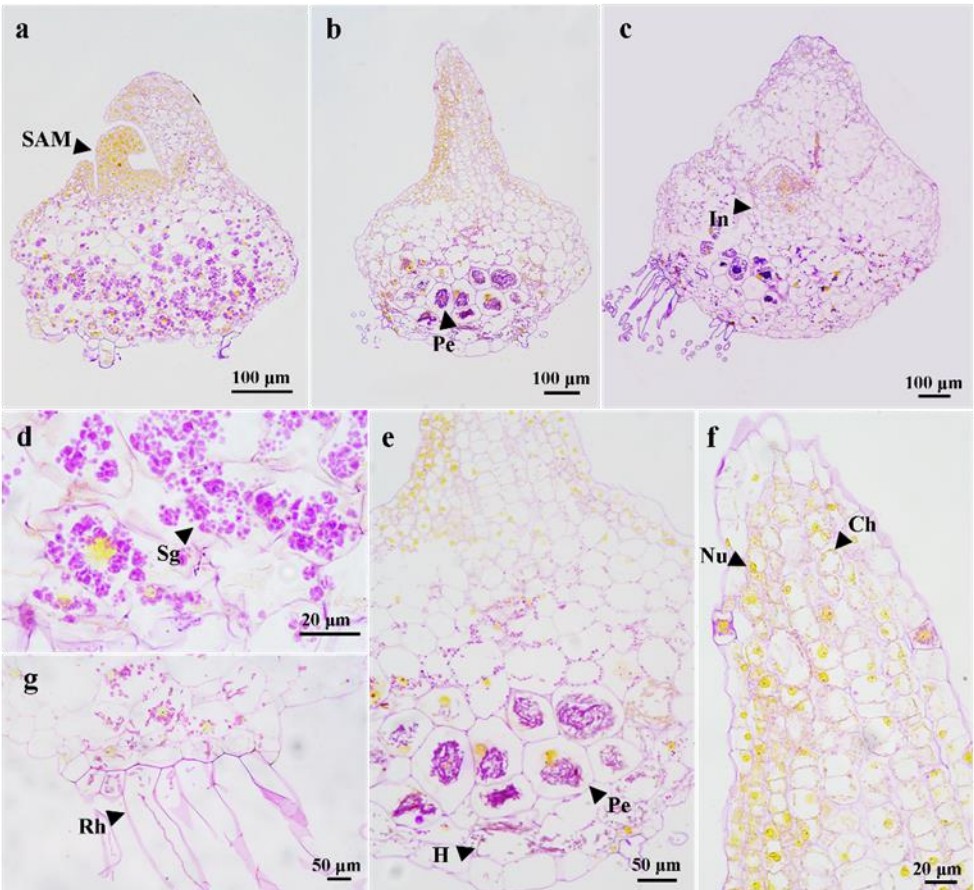

**Figure 6.** Longitudinal sections of symbiotically and asymbiotically germinated protocorms of *Dendrobium chrysotoxum* stained with PAS and Naphthol yellow S. (**a**) Asymbiotic protocorm of stage 4 with well-developed SAM, a large amount of starch granules stained purplish red, and nuclei stained yellow; (**b**) Symbiotic protocorm of stage 4, pelotons (Pe) stained purplish red, and nuclei stained yellow; (**c**) SAM initials (In) with conspicuous nuclei appearing at the center of symbiotic protocorms; (**d**) Starch granules (Sg) in asymbiotic protocorms; (**e**) Pelotons (Pe) in the inner cortex cell and hyphae (H) in the outer cortex cell in symbiotic protocorms; (**f**) The mesophyll cell with chloroplasts (Ch) in the cotyledon and nucleus (Nu) stained yellow; (**g**) Rhizoids (Rh) in symbiotic protocorms.

At stage 5, a furrow appeared on one side at the base of the cotyledon by 40 d (Figure 5e), then the first visible true leaf emerged from this furrow (Figures 3e and 5f,g). Histological sectioning revealed that an adventitious root primordium formed near the SAM (Figure 4j). We consider this stage the beginning of the seedling stage. At stage 6, roots subsequently emerged at the base of the protocorm, and the second true leaf emerged (Figure 3f). In this stage, *D. chrysotoxum* seedlings formed, with well-developed roots and leaves and a persistent cotyledon (Figure 3f–h). Additionally, the fungus had colonized the roots, and a mycorrhizal association formed with *D. chrysotoxum* seedlings (Figure 4k).

### 3.3. Morphological Changes of Pelotons in Symbiotically Germinated Protocorms

Using anatomical technologies and SEM, we observed the entry and colonization of hyphae GC-15 in protocorm cells during seed germination. Hyphae and pelotons could be easily stained by hematoxylin, PAS, and Trypan blue. When the seeds of *D. chrysotoxum* were swollen via water absorption within 1 week, the hyphae invaded seeds through the end of the seed micropyle (Figure 4l). Pelotons were formed in some large parenchymatous cells in the lower part of protocorms at stage 2 (Figure 4b,m). Then, at stage 3, as numerous rhizoids formed, numerous hyphae entered protocorm cells through rhizoids (Figure 4n).

After penetrating the protocorm cells through rhizoids, dispersed hyphae but no pelotons were present in outer cortex cells, whereas characteristic pelotons appeared in inner cortex cells. The upper part of protocorms and protrusions harbour neither hyphae nor pelotons. The pelotons initially formed loose hyphae coils, which were healthy and alive (Figure 5h). The intercellular hyphae were visible, and they penetrated the cortical cell wall and entered the adjoining cortical cell (Figure 5k). Hyphae degradation occurred subsequently. Some pelotons became tightly entangled, forming tight clumps that were eventually digested into hyphal fragments, following which they disappeared (Figure 5i,j,l). Both collapsed hyphae and healthy pelotons could be observed during protocorm development (Figure 6e).

### 3.4. Comparative Symbiotic and Asymbiotic Seed Germination

The seed germination rate and the effectiveness of symbiotic seed germination (cocultured with GC-15 on OMA medium) and asymbiotic germination (cultured on 1/2 MS medium alone) were evaluated using the seed germination stages defined from the descriptions above (Figure 3, Figure 4 and Figure S1). Germination was defined as the swollen embryo appearing (stage 1). The germination percentage of seeds cocultured with GC-15 was $93.00 \pm 2.58\%$ by 15 d and reached 100% by 45 d. While on 1/2 MS medium, the germination percentage was $79.99 \pm 3.24\%$ and reached 100% by 60 d. After 15 d of culturing, seeds in symbiotic and asymbiotic cultures were swollen and developed into stage 2 at similar rates of 88% and 76%, respectively (Figure 7a). By 30 d, seeds in symbiotic and asymbiotic cultures had developed into stage 3 at a rate 67% and 77%, respectively (Figure 7b). By 45 d, seeds in symbiotic culture developed into stage 4 very soon at a rate of 55.5%, but most seeds (75.5%) in asymbiotic culture developed into stage 3 and no seeds developed into stage 4 (Figure 7c). By 60 d, the percentages of seedlings (stage 6) in asymbiotic culture were significantly high, compared to a very low percentage of seedling formation in symbiotic culture (Figure 7d). By 75 d, the percentages of seedlings in asymbiotic culture were also significantly higher than those in symbiotic culture (Figure 7e).

Morphologically, the process of symbiotic and asymbiotic germination was similar and experienced six stages (Figures 3 and S1). However, subtle morphological and developmental differences were observed between the two germination methods. A significant difference was observed at stage 3 when the symbiotic protocorm developed numerous rhizoids, which were long and thin, white in color, and grew radially at the base of the protocorm (Figures 3c and 6g). On the contrary, the rhizoids that grew randomly on the base of the asymbiotic protocorm were transparent, stumpy, and rare (Figure S1c).

Both symbiotic and asymbiotic protocorms at stage 4 were stained with histochems either using the PAS reaction or Naphthol yellow S. Histological analysis showed that pelotons were absent in asymbiotically germinated protocorms, whereas many pelotons were observed in symbiotically germinated protocorms (Figure 6a–c). The PAS staining showed that a large number of starch granules were detected in the main body of the aymbiotically germinated protocorms. In contrast, there were also abundant polysaccharides in some large parenchyma cells at the basal part of symbiotically germinated protocorms; however, those polysaccharides were apparently hyphae and pelotons in shape. There were only a few starch granules present in the upper cells of protocorms. Both in asymbiotic and symbiotic protocorms, nuclei, especially in meristematic cells, were strongly Naphthol yellow S positive, but no other apparent protein particles were detected.

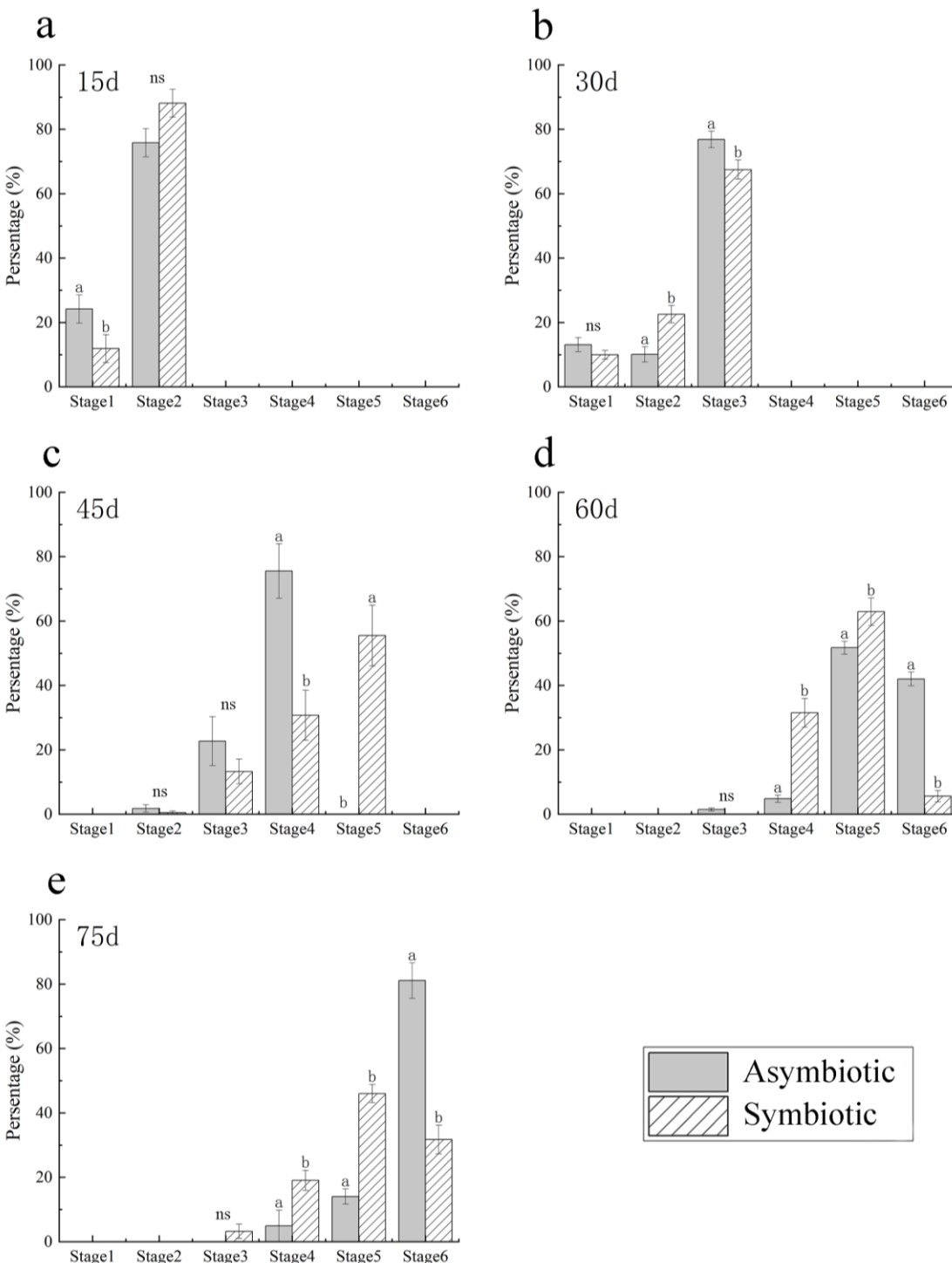

**Figure 7.** Comparative effects of symbiotic and asymbiotic cultures on protocorm development stages during seed germination for 15 d (**a**), 30 d (**b**), 45 d (**c**), 60 d (**d**), and 75 d (**e**). The percentage represents the mean ± SE of five repeated experiments. Treatments with different letters are significantly different, and "ns" is not significant within each stage at α = 0.05.

## 4. Discussion

### 4.1. The Structure and Function of Protocorms

The majority of orchid seeds have limited differentiation in their embryos without cotyledons and endosperm, but they germinate into protocorms before becoming seedlings [2,34]. The protocorm is the transitional structure from an embryo to a seedling.

The protocorm is an important stage in the orchid life cycle and is designed to house its symbiont and form a shoot apical meristem [7]. However, orchid protocorm properties and functions have been controversial for a long time, and no agreement has been reached.

The protocorm development has been morphologically categorized into 5–7 stages in various terrestrial and epiphytic orchid species [17,19,35–38]. The general morphological changes include embryo enlargement, foliaceous organs, and root appearance. The anatomical structure of the protocorm changes during seed germination, however, and remains poorly understood. In this study, we investigated the protocorm development process of *D. chrysotoxum* co-cultured with its mycobiont, *Tulasnella* sp., and characterized the protocorm development into six stages based on both morphological and anatomical features (Figure 8). The general morphology of the protocorm development stages of *D. chrysotoxum* is similar to those described for other *Dendrobium members* [20,21,24,39,40]. In addition, a detailed description of anatomical changes is also presented in our study (Figure 8): Swollen embryos are primarily the result of large cells absorbing water at stage one; the protomeristem initiates at stage two and then develops into the SAM initial at stage three; the SAM forms from the SAM initial and then develops into leaf primordia at stage four.

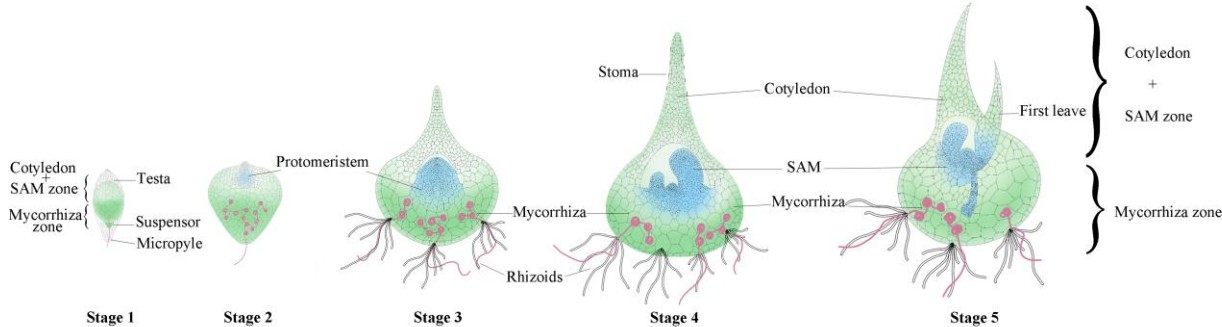

**Figure 8.** Schematic illustration for developmental stages of the *Dendrobium chrysotoxum* protocorm showing morphogenesis changes in protocorm development during symbiotic germination.

Orchid protocorms are formed after embryos are swollen and the testa rupture. The protocorms are generally globular or spherical in shape, green or cream yellow in colour, with a foliaceous organ on the top [7,12]. Due to a poor understanding of SAM development, this foliaceous organ at the apical of the protocorm has yet to be identified as a leaf or a cotyledon [7]. Usually, a protrusion or dorsal crest structure forms on the flat upper surface of orchid protocorms, which is considered a protomeristem or SAM, and subsequently elongates and develops into the first leaf [1,16,17,41]. Several authors have referred to this organ as a cotyledon due to its initiation and vascular structure, which are similar to those of monocots [21,22,42]. Some authors suggest that several terrestrial orchids, e.g., *Bletilla* spp., *Sobralia macrantha*, *Arundina graminifolia*, have a rudimentary cotyledon in their seeds, whereas most orchid species have no cotyledon [1,41]. Nishimura [42] reported that the most developed cotyledon can be found in *Bletilla striata*, in which the cotyledon becomes a leaf-like structure with a vascular bundle. Batygina et al. [15] pointed out that this organ should be referred to as the shoot apex rather than a cotyledon or a leaf. In *D. chrysotoxum*, the embryo develops into an oblate spheroid protocorm with a protrusion at stage two, and then this protrusion develops into a foliaceous organ with well-developed stomata and chloroplasts at stage four. In seed plants, the cotyledon is the primary structural component of the mature embryo, which arises directly from embryonic tissue, not the stem apical meristem [43]. Our histological analysis showed that this protrusion differentiated from embryonic tissue at the apical part of the *D. chrysotoxum* protocorm but not from SAM. Therefore, the protrusion could be considered a cotyledon rather than the first leaf, similar to some epiphytic species [21,44]. However, in this study, the SAM initials apparently developed after the appearance of the protrusion and were located beneath the protrusion (Figure 4c). Moreover, our results also indicate that SAM originated from embryonic

tissue in the protocorm and subsequently developed into a fully developed SAM with leaf primordium (Figure 4b–i).

Cotyledons are primarily responsible for providing nutrients for seed germination and seedling establishment in seed plants. In *D. chrysotoxum* protocorm, the cotyledon has structural characteristics similar to those of mesophyll tissues in seed plants. For instance, stomata are present on the epidermis, and inner parenchyma cells contain numerous chloroplasts. The anatomical features of the cotyledon indicate that *D. chrysotoxum* protocorms have the capacity for photosynthesis and may function as photosynthetic organs during seed germination. Several epiphytic orchids, such as *Dendrobium*, *Vanda*, and *Dendrophylax*, have been documented to have protocorm photosynthetic ability [16,17,45,46]. Hew and Khoo [45] indicated that *Dendrobium taurinum* protocorms resemble adult leaves in their response to various environmental factors and have the capacity for crassulacean acid metabolism. The capacity of photosynthesis in cotyledons during protocorm development might be an adaptation for epiphytic habit. Epiphytic orchid seeds germinate directly on the tree, and protocorms turn green soon and perform photosynthesis until the true leaves appear. Consequently, more nutrients will be provided to the protocorm, resulting in faster growth and root emergence to attach to the host tree surface [17,24,47].

Unlike other seed plants, orchid seeds lack endosperm, shoot apical meristems, and cotyledons; however, these structures are gradually formed during protocorm development. Therefore, some researchers consider orchid seeds developmentally immature, but protocorms differentiate from germinated embryos, which are mature after germination [2,13]. The development of protocorms is actually a process in which the embryo continues to develop and mature after seed germination [2]. Alternatively, some researchers regard protocorms as underdeveloped seedlings because protocorms have SAM and root primordia and are capable of photosynthesis [15–17]. Hoang et al. [17] propose that protocorms are initially the continuations of embryos, and then after forming the leaf primordium, they enter seedling development. Similar to other orchid species, a polarized embryo of *D. chrysotoxum* is divided into two zones: a small cell zone in the upper part and a large cell zone in the lower part (Figure 8). The small cell zone will give rise to a cotyledon and a SAM located on the upper part of the protocorms subsequently. Therefore, the protocorm can be regarded as a continuation of the embryo in some respects. The large cell zone will give rise to mycorrhiza, which is responsible for the fungal association. Yeung [6,7] suggested that cell fate determination has already been completed in orchid embryos and that the protocorm is a stage of development based on the embryo's "blueprint".

### 4.2. Associated Fungi Colonization and Digestion during Protocorm Development

The germination of orchid seeds is highly dependent on obligate fungi [48]. The unique properties of protocorms may provide an easy entry for fungal invasion and colonization [12,49–51]. The site of hyphal entry may be important in establishing mycorrhizal associations. It is possible that structural and physiological differences in protocorm development can determine which symbiont is best suited for a particular orchid [7]. When seeds of *D. chrysotoxum* were co-cultured with *Tulasnella* sp., a fungal invasion was observed when the embryo absorbed water (Figure 4i). The fungal hyphae initially enter from the micropyle of seeds and then infect the embryo's basal degenerate suspensor. Suspensors generally degenerate when the orchid embryo matures and are composed of large, elongated, dead cells with weak or absent cuticles that may reduce fungi colonization resistance [1,52–54]. Degenerated suspensors appear to be the only entry site for the fungal hyphae to invade protocorms, resulting in a successful mycorrhizal association [51,55]. However, Rasmussen et al. [49] observed in *Dactylorhiza majalis* that a compatible fungus infected the suspensor and did not form pelotons, but the hyphae that entered through rhizoids can successfully establish a mycorrhizal association. In *D. chrysotoxum*, numerous rhizoid clusters formed from aggregate epidermal cells and appear at the base of the protocorm at stage three. It is also possible for hyphae to penetrate through rhizoids during protocorm formation [49,51,56]. Trypan blue staining and SEM observation clearly

show that the hyphae penetrated the rhizoids in *D. chrysotoxum* protocorms (Figure 4n), indicating that the rhizoids are the ideal location for hyphae entry. Mycorrhizal fungi can produce cellulase and pectinase, allowing them to easily enter the orchid's protocorm [57]. Furthermore, the pressure generated by the hyphae's cytoplasm may also allow them to penetrate protocorm cells [22]. In contrast to symbiotically germinated protocorms, fewer and shorter rhizoids appeared on the asymbiotically germinated protocorms of *D. chrysotoxum*. Accordingly, mycobiont fungi may be involved in the morphogenesis of the orchid's protocorm.

Obtaining nutrients from pelotons is an essential strategy in orchids for seed germination and successful seedling development. Interestingly, the fungi will ultimately be digested by orchid protocorm cells after peloton formation. The fungal colonization in *D. chrysotoxum* first occurs when the embryo swells, enlarges, and forms pelotons within inner cortex cells at stage two of germination (Figure 4b). Our study clearly demonstrates the process from fungal colonization to digestion in *D. chrysotoxum*. First, the fungal hyphae penetrate rhizoids and colonize the inner cortex of the protocorm; then, they develop from loose pelotons to dense ones, which are digested by protocorm cells for nutrients (Figure 5h–j). It is impossible for hyphae to colonize any cell of the protocorm. As such, peloton formation is always restricted to the large parenchyma cells of the inner cortex (Figure 6b,e). It might be due to an abundance of hydroxyproline-rich glycoproteins in the cell walls of the inner cortex [58]. Using TEM, morphological changes of fungi during colonization and digestion were examined in several orchid species, such as *Spiranthes sinensis*, *Bletilla striata*, *Dendrobium officinale*, etc. [22–24]. Hyphae penetrate cortex cell walls by the pressure generated from the cytoplasm to infect adjacent cortical cells and form a new peloton [22,24,38]. A large number of hydrolytic enzymes are produced in the plasma membrane, and the membrane is invaginated and encloses the peloton, which can be digested [22,24,58–60].

In orchid seeds, protein and lipids are stored, but starch is absent [9]. Therefore, any reserve carbohydrates should be obtained from an external source for seed germination [2]. Fungal pelotons are digested and provide nutrients to orchids during seed germination [5,50]. It is proposed that orchids receive carbohydrates from the fungus and sometimes amino acids (glutamine, glutamic acid, aspartic acid), and nicotinic acid [61,62]. In our study, both alive and digested pelotons can be stained by PAS within symbiotically germinated protocorms of *D. chrysotoxum* (Figure 6b,e), indicating that pelotons are primarily composed of polysaccharides, which are available carbon sources for orchids. Previous studies found that digested hyphae have thickened cell walls without cytoplasm [22,63,64]. Consequently, the main components of pelotons digested by orchids are carbohydrates, which are derived from the cell walls of fungi, such as cellulose and chitin. However, asymbiotically germinated protocorms obtain nutrients from artificial media and produce large quantities of starch granules for subsequent protocorm development. Our results indicate that carbohydrates from mycobiont play a crucial role in seed germination and protocorm development of *D. chrysotoxum*. In *D. chrysotoxum*, seed germination rates in symbiotic cultures are higher than those on asymbiotic culture with the most optimal artificial medium. The result indicates that seeds of *D. chrysotoxum* can successfully germinate and develop into seedlings by using nutrients provided by only a single compatible fungus. However, symbiotic germination was slightly slower in terms of seedling formation than asymbiotic germination (Figure 7). This result indicates that there may be a conversion of fungi in *Dendrobium* seedlings, and more fungi need to be recruited to obtain sufficient nutrition for subsequent seedling development [65].

## 5. Conclusions

In this study, we clarified the process of protocorm development as well as the structure and properties of embryos and protocorms in *D. chrysotoxum*. There are six developmental stages that were morphologically and anatomically defined during symbiotic seed germination (Figure 8). Embryos of *D. chrysotoxum* develop polarized cell regions whose

developmental consequences are programmed. Small embryonic cells at the top develop into cotyledons and SAM, and large embryonic cells at the base develop into rhizoids for fungal hyphae accessibility, and special parenchyma cells as a symbiotic place for fungi to colonize (Figure 8). The protocorm of *D. chrysotoxum* might consist of a cotyledon, SAM, mycorrhiza, and rhizoids, each of which has its own functions, e.g., cotyledons act as photosynthesis organs, shoot apical meristems lead to seedling formation, rhizoids serve as entry points for fungal invasion, and mycorrhiza provide nutrients for growth.

**Supplementary Materials:** The following supporting information can be downloaded at: https://www.mdpi.com/article/10.3390/horticulturae9050531/s1, Figure S1: Developmental stages of asymbiotic seed germination of *Dendrobium chrysotoxum.* (a) Stage 1 Embryo (Em) swollen; (b) stage 2 Embryo enlargement and testa (T) ruptured; (c) Stage 3 appearance of cotyledon (Co) and rhizoids (Rh); (d) Stage 4 continued elongation of cotyledon (Co); (e,f) Stage 5, appearance of the first (FL) and root (R); (g–i) Stage 6 plantlet with cotyledon (Co), true leaves and root. FL: First leaf; SL: Second leaf.

**Author Contributions:** Conceptualization, Y.L. (Yan Luo) and S.S.; methodology, Y.L. (Yan Luo) and S.S.; validation, Y.L. (Yan Luo), X.G. and Y.W.; formal analysis, Y.L. (Yan Luo), X.G. and D.D.; investigation, X.G., D.D. and Y.W.; resources, Y.L. (Yan Luo) and S.S.; data curation, Y.L. (Yan Luo) and X.G.; writing—original draft preparation, Y.L. (Yan Luo) and X.G.; writing—review and editing, S.S. and Y.L. (Yinling Luo); visualization, Y.L. (Yan Luo), S.S. and X.G.; supervision, Y.L. (Yan Luo). All authors have read and agreed to the published version of the manuscript.

**Funding:** This research was funded by the National Natural Science Foundation of China, grant numbers 32270225 and 32171655, the West Light Talent Program of the Chinese Academy of Sciences, grant number E1XB011B01, and the Southeast Asia Biodiversity Research Institute, Chinese Academy of Sciences, grant number Y4ZK111B01.

**Institutional Review Board Statement:** Not applicable.

**Informed Consent Statement:** Not applicable.

**Data Availability Statement:** The data presented in this study are available upon request from the corresponding author.

**Acknowledgments:** We thank Shibao Zhang of the Institute of Botany, Chinese Academy of Sciences, and Baohua Song of the University of North Carolina at Charlotte for valuable comments on a preliminary version of the manuscript. We thank Sven Landrein of Kadoorie Farm and Botanic Garden, and Ping Yates of the University of North Carolina at Charlotte for their help with English writing. We thank Lu Tang, and Peng Sun from the Xishuangbanna Tropical Botanical Garden, Chinese Academy of Sciences for their help in experiment data analysis. We are grateful to Ting Tang of the Public Technology Service Center, Xishuangbanna Tropical Botanical Garden, Chinese Academy of Sciences, for technical assistance with SEM images.

**Conflicts of Interest:** The authors declared no conflict of interest.

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
