# Peer review of "Morphogenesis Changes in Protocorm Development during Symbiotic Seed Germination of Dendrobium chrysotoxum (Orchidaceae) with Its Mycobiont, Tulasnella sp."

_horticulturae, doi:10.3390/horticulturae9050531_

Round 1

Reviewer 1 Report

The manuscript presents a microstructural and anatomical description of the germination process of Dendrobium chrysotoxum and the effect of inoculation with Tulasnella sp. The results indicate a significant influence of the association with the mycorrhiza for the formation of structures and initial development.

The work is only descriptive, the presentation of the results and the discussion of these are based on representative observations of the germination process. It would be convenient to mention in the results if the anatomical studies of microstructure and association with Tulasnella sp, are the result of a significant number of replicates with reproducible results.

I suggest that the objective of the work is limited only to the morphological and anatomical development of Dendrobium chrysotoxum seeds, since mentions are made of the physiological and molecular effects, however, the experimental design does not include any experiment that allows discussing these aspects.

The conclusion must be completely rewritten, since the conclusions of the work are not clearly and precisely mentioned, and very general aspects of the introduction and the ecological and evolutionary implications of the study are mentioned.

Additional suggestions

Page 3, Line 127. Change "PH" to "pH"

Reviewer 2 Report

1. This manuscript provides detailed developmental pathways of symbiotic germinated seeds of an orchid, Dendrobium chrysotoxum, using light and scanning electron microscope, which can be fundamental information for orchid researchers to understand the influence of orchid mycorrhizae on the germination process and subsequent development.

2. It is original research and has addressed the detailed developmental pathway of symbiotic seeds of orchids by providing photos taken by a scanning electron microscope, which is rare to be seen in this kind of research. 3. They used a Dendrobium orchid, which is important for medicinal usage and also ornamental value.

4. They need more quantitative data together with statistics for present differences in growth and development between symbiotic and symbiotic seeds.

5. The conclusions consistent with the evidence and arguments presented and do they address the main question posed.

6. The references appropriate.

7. All figures look good.

8. Orchid embryos are undifferentiation, so they only consist of anterior and posterior zones, after germinating, they develop into protocorms, and then seedlings. I don't understand why a protocorm has a cotyledon at stage 4. Based on my knowledge, that is a scale leaf.

9. What is the difference, such as morphology and growth and development, between asymbiotc vs symbiotic seeds? The authors should compare them more systematically.

Round 2

Reviewer 1 Report

The authors made the suggested corrections

Author Response

Dear reviewer,

Thank you for your valuable suggestions. We have checked the references list, and make sure that all references are relevant to the contents of the manuscript. The English writing has also been checked thoroughly, and some minor English grammar errors have been corrected.